# A profession in danger: Stakeholders' perspectives on supporting the pharmacy profession in Lebanon

**Mohamad Alameddine**[1,2], **Karen Bou-Karroum**[1], **Sara Kassas**[1], **Mohamad Ali Hijazi**[3]*

**1** Department of Health Management and Policy, Faculty of Health Sciences, American University of Beirut, Riad El Solh, Beirut, Lebanon, **2** College of Medicine, Mohammed Bin Rashid University of Medicine and Health Sciences, Dubai Health Care City, Dubai, United Arab Emirates, **3** Department of Pharmaceutical Sciences, Faculty of Pharmacy, Beirut Arab University, Beirut, Lebanon

* m.hijazi@bau.edu.lb

## Abstract

### Background

Lebanon boosts one of the highest pharmacists to population ratios globally (20.3/10,000). Yet, workforce analysis elicited serious concerns with the distribution, practice environments and regulation of Lebanese pharmacists. Recent workforce data shows that the profession has been majorly destabilized with hundreds of pharmacists closing their pharmacies or losing their employment. Proper planning for the future of the pharmacy profession in Lebanon necessitates a deeper understanding of the current challenges and the necessary policy and practice recommendations. The aim of this study is to examine stakeholders' perspectives on the current pharmacist workforce challenges and the necessary measures to support the profession.

### Methods

The research team carried out a series of semi-structured interviews with twenty-one key stakeholders within the pharmacy profession in Lebanon. We categorized stakeholders according to their experience as policy makers, practitioners, academicians, and media experts. The interview guide included questions about workforce trends, labor market challenges and recommendations for improvement. Interviews were transcribed and analyzed thematically.

### Results

Four major themes emerged from this study: the oversupply of pharmacists in Lebanon, the demand supply imbalance, poor regulation of the pharmacy practice, and the difficult practice environment. There was a consensus among interviewees that the oversupply of pharmacists is due to the poor workforce planning and weak regulatory framework, combined with the easy integration of foreign-trained pharmacists into the labor market. The lack of coordination between the educational and practice sectors is further widening the demand-supply gap. Interviewees further revealed that the regulatory policies on pharmacy practice

**Data Availability Statement:** Data cannot be shared publicly because it contains potentially identifying information of human subjects. Data are available from the American University of Beirut

Institutional Data Access/Ethics Committee (contact via email: irb@aub.edu.lb) for researchers who meet the criteria for access to confidential data.

**Funding:** This study was funded by the University Research Board of the American University of Beirut, received by MA. Award number 103785-Project number 25278.

**Competing interests:** The authors have declared that no competing interests exist.

were outdated and/or weakly enforced which increases the risk of unethical practices and erodes the image of pharmacists in the society. With respect to the practice environment, there is an ongoing struggle by Lebanese pharmacists to maintain profitability and exercise their full scope of practice.

## Conclusion

The poor pharmacy workforce planning and regulation is significantly weakening the pharmacy profession in Lebanon. A concerted effort between the various stakeholders is necessary to enhance workforce planning, regulate supply, optimize the integration of pharmacists into work sectors of need, and improve the financial and professional wellbeing of pharmacists in Lebanon.

## Introduction

In the last several decades, the pharmacy profession has shifted from a product-oriented profession to a patient-oriented one [1]. Nowadays, pharmacists are expected to function as caregivers, decision-makers, communicators, managers, life-long learners, teachers, leaders and researchers [2]. With the rapidly aging population and increased usage of medicines, the demand for medication experts has increased [3]. These changes contributed to the increased student enrollment and expansion of pharmacy schools [1]. Several countries including the United States and the United Kingdom, were concerned that this rapid expansion might have detrimental effects on the future workforce. Similar apprehensions have emerged in Lebanon (an Eastern Mediterranean country), which has seen an expanding market of pharmacy schools and an increasing number of pharmacy graduates [4].

Countries all over the world regularly assess the number of pharmacists needed in response to population density and market demands. Such assessments are necessary to avoid workforce imbalances (oversupply or undersupply). The oversupply of healthcare professionals is associated with the risk of unemployment and/or precarious work. Precarious work leads to dissatisfaction and consequently, to poor mental health and burnout [4]. In addition, the oversupply of pharmacists results in reduced salaries and increased workforce casualization as the demand decreases [5]. As such, many countries have implemented policy measures to limit the number of pharmacy students [1].

### The pharmacy workforce local context

Lebanon is a small Eastern Mediterranean developing country with an estimated population of around five million citizens added to an estimated two million Syrian and Palestinian refugees [6]. According to the Order of Pharmacists of Lebanon (OPL), the Lebanese population is served by 8,800 pharmacists, registered with the OPL. Pharmacy students can graduate with a bachelor's in pharmacy degree (5-year pharmacy program) or a PharmD. degree (6-year pharmacy program). There are five universities in Lebanon that have pharmacy programs, with a number of private universities seeking approval to open pharmacy schools from the Ministry of Education and Higher Education (MEHE). Lebanese pharmacy programs graduate 190–240 pharmacists per year [4]. Due to the absence of a national plan guiding the supply and utilization of pharmacists, there is no accurate database on the exact distribution of degrees awarded between the B.S. and the Pharm.D.

A recent examination of the pharmacist workforce trends and distribution in Lebanon revealed a significantly high pharmacist to population ratio of 20.3 pharmacists per 10,000 population. The study also indicated a high ratio of pharmacies to population, resulting in limited profitability and dissatisfaction among community pharmacists, coupled with negative influences on their professional status. Despite the oversupply of pharmacists, Lebanon has a modest number of hospital pharmacists. In fact, the hospital pharmacists' role is limited to dispensing medicines, compounding, and cost management rather than practicing the clinical role they have been trained to serve [7]. Workforce analysis also reveals a low ratio of community pharmacists to pharmacies in Lebanon, which raises concerns over the workload, working hours, and the quality of services delivered [7].

Over the last few months, Lebanon's worsening economic crisis combined with anti-government protests and the COVID-19 pandemic, have had dire consequences on the pharmacy profession. The OPL confirmed that the number of closed pharmacies exceeded 200, and about 1000 pharmacies are at risk of shutting down due to the reduction in prices of pharmaceutical products and the reduced purchasing capacity of citizens [8, 9]. Lebanon's pharmaceutical importers are also struggling with the economic meltdown and currency depreciation and are voicing concern with an imminent shortage in the supply of essential drugs [10]. Under such circumstances, it comes as no surprise that the incidences of pharmacy theft have seen a remarkable upsurge adding to the woes of pharmacists in Lebanon [11].

Proper planning for the future of the pharmacy profession in Lebanon necessitates a deeper understanding of the current challenges and the necessary policy and practice recommendations to improve the workforce planning and working conditions. Recent studies have provided a better understanding of the numbers and distribution of pharmacists in Lebanon, yet a more comprehensive understanding of the specific workforce issues and their underlying causes is still lacking. The aim of this study is to examine pharmacy stakeholders' perspectives on the current pharmacist workforce challenges and the necessary measures to support the profession.

## Methods

### Sampling and data collection

This research employed a qualitative design utilizing semi-structured key informant interviews. A number of key stakeholders were identified to help understand workforce trends, describe workforce challenges, and suggest corrective actions. Interviewees were invited to participate based on their knowledge of the field and engagement in the supply, regulation and/or organization of the profession. A review of public records was carried out to identify a number of key stakeholders in the public domain, those constituted the seed list and were interviewed first [12]. Snowball sampling was utilized during the interviews and the interviewees were asked to underscore the names of other experts whom they recommend the research team to interview [13]. The research team kept interviewing until there were no more experts to recommend and no new themes were emerging [14]. At the end of the study, twenty-one diverse stakeholders within the profession were invited to participate. The variability of interviewees across geographical locations (e.g. across the six Governorates of Lebanon) and institutions was ensured. They were categorized according to their work experience under policy-makers, practitioners, academicians, and media (Fig 1).

The majority of the interviewees had experience in policy-making. They were divided into those who had experience in the Order of Pharmacists in Lebanon (OPL), Ministry of Public Health (MOPH), Parliamentary Health Committee, and Syndicate of Hospitals in Lebanon. All policy makers had extensive experience in the pharmacy labor market in Lebanon (owner of community pharmacies, work at or own a pharmaceutical company, working at

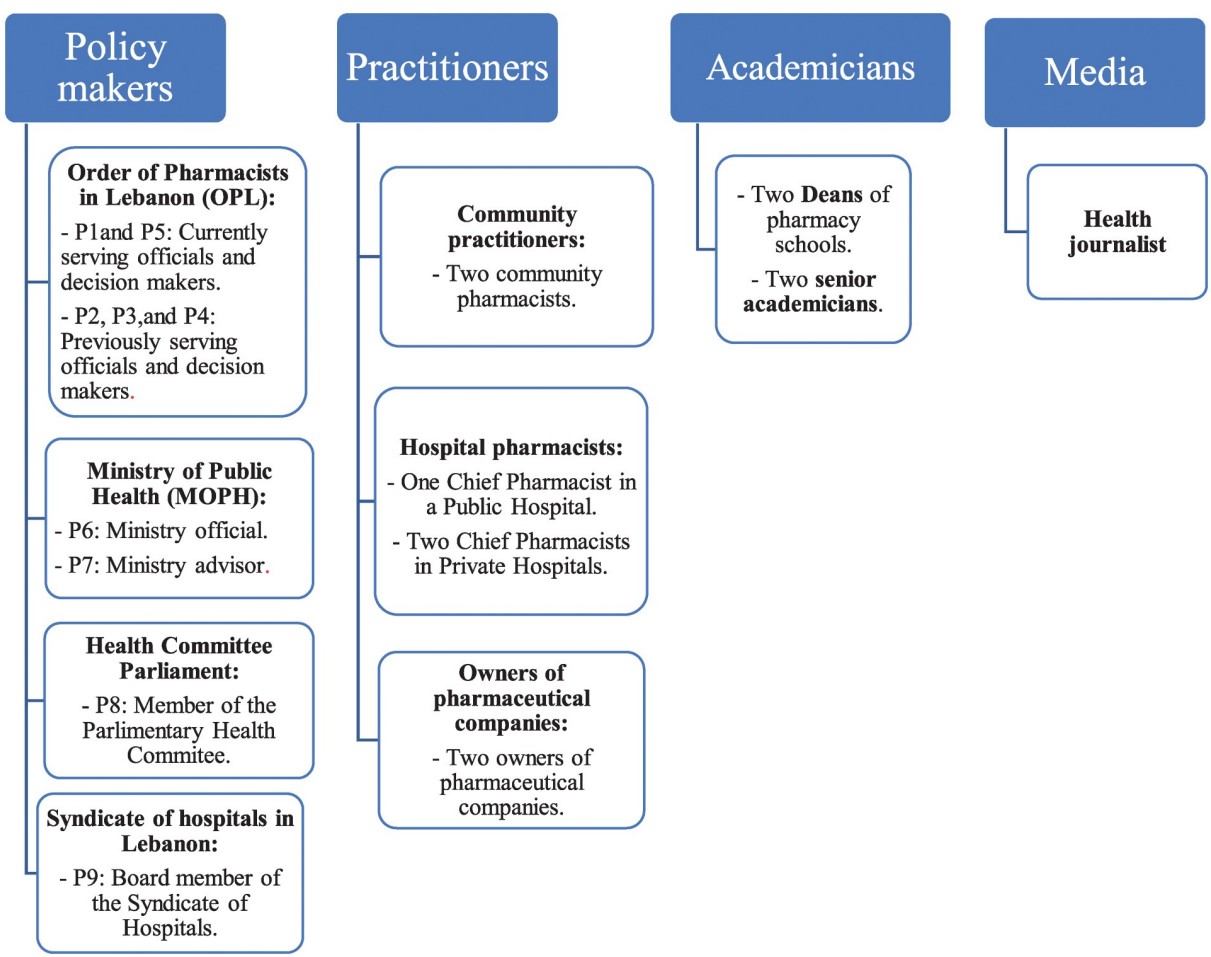

**Fig 1. Categorization of interviewees.**

pharmaceutical industries, etc.). Interviewed practitioners included two community pharmacists, three chief hospital pharmacists, and two owners of pharmaceutical industries. They were all registered pharmacists from various areas of the country. One of the community pharmacists was also a journalist and an advocate of human rights. The remaining interviewees were one health journalist and four academicians, including two key decision makers/Deans of Faculties of Pharmacy, and two Professors of Pharmacy at local universities.

The research team invited each of those interviewees via email. Upon the receipt of the consent to participate in the study, the interview was scheduled at a time and place of convenience to the interviewee. The research team shared a recently published workforce analysis with the interviewees before conducting the interviews to provide them with the latest evidence. By the end of the study, a total of 21 single face-to-face semi-structured interviews were conducted. All interviews were digitally recorded (with the interviewees consent) and lasted between 15 and 40 minutes. Interviews continued until the saturation point was reached no new information was identified [14].

The interview guide was developed based on existing literature and the objectives of the study. It included questions about the interviewees' perspectives on workforce trends, labor market challenges and recommendations for improvement. Probing questions were asked to gain a better understanding of interviewees' views and comments [15]. As such, interviewers

asked clarifying questions such as "why do you think this is the case", "what do you think is at the root cause of this problem?". Below is a copy of the interview schedule used in this study:

1. Please describe your experiences in the pharmacy sector in Lebanon?

2. What is your opinion on the labor market of Lebanese pharmacists?

   - What needs to be improved?

   - Why do you think this is the case?

3. What necessary measures need to be implemented to ensure future balance in the pharmacy labor market in Lebanon?

   - Why would these measures be important?

   - How could they be implemented?

4. What are the requirements to upscale the role of pharmacist in Lebanon?

   - What are the impediments?

   - What needs to be done to overcome them?

   - Who are the responsible parties?

5. How do you describe the practice environment of pharmacists in Lebanon?

   - Why do you think this is the case?

   - How do you think this could be improved?

6. Are there any additional issues or improvements you wish to highlight?

7. Who else should we speak with to help answer our questions?

## Data analysis

The interviews conducted were fully transcribed. Transcribed interviews were comprehensively read to gain preliminary insight into the data. Transcripts were then imported to Dedoose, a qualitative analysis software [16]. All transcripts were read thoroughly by two members of the research team who met regularly to resolve disagreements in qualitative coding. Disagreements were discussed until consensus was reached. We used the grounded theory approach in our analysis, a process of coding that uses phrases and terms used by interviewees for developing codes [17]. Two members of the research team completed inductively, line-by-line coding for the transcripts without the use of a priori coding scheme. Codes were attached to excerpts, and an initial set of codes was developed from the recurring ideas and discussion points. Multiple rounds of co-coding were applied until consensus on codes definitions was reached. Duplicate codes were removed and overlapping codes were merged. Then, the identified set of codes was explored to form sub-themes from the conceptually related codes. Finally, sub-themes were grouped to form overarching themes. The themes were applied to each transcript, with amendments made if new codes emerged. The final list of themes and sub-themes was approved by all research team members [18].

## Ethical considerations

This study was approved by the Institutional Review Board of both the American University of Beirut (Protocol number SBS-2019-0103) and Beirut Arab University (Protocol number

2019H-0060-P-R-0301). A written or oral informed consent was obtained from all interviewees. Confidentiality and anonymity of interviewees were guaranteed. Interviewees had the right to refrain from answering any questions and to end the interview at any time during the interview.

## Results

The thematic analysis of interview data generated four major themes and eleven associated subthemes (Table 1).

### Contributing factors to the oversupply of pharmacists in Lebanon

**Poor regulation of supply.** Many interviewees acknowledged that the high number of pharmacy graduates was due to the lack of adequate assessment of the number of pharmacists needed relative to the total population number. They mentioned that the MOPH, OPL, and MEHE are responsible for conducting these assessments. Several interviewees referred to this restriction on the number of pharmacy graduates as "numerus clauses", used by governments in several other countries:

*"There is something called 'numerus clauses' that other countries apply. It specifies the number of pharmacists that the population needs. . .and accordingly there is an exam that places criteria on who is to be accepted into pharmacy."*

(*Policy Maker 6*)

**No barriers to the integration of foreign trained pharmacists to Lebanon.** Several interviewees agreed that the number of graduates inside the country and those coming from abroad were contributing to the increased number of pharmacists in Lebanon. They noted that pharmacists who studied abroad were encouraged to practice in Lebanon due to the ease of integration into the labor market. All that is required is just to pass a simple written and oral exam. Interviewees highlighted the importance of limiting the return of pharmacists from abroad

**Table 1. Overview of the main themes that emerged from the qualitative analysis.**

| Themes | Sub-themes |
|---|---|
| 1. Contributing factors to the oversupply of pharmacists in Lebanon. | • Poor regulation of supply.<br>• No barriers to the integration of foreign trained pharmacists to Lebanon.<br>• Public perception of the pharmacy profession. |
| 2. The demand-supply imbalance in the pharmacist workforce. | • Need for additional investments in the pharmaceutical sector.<br>• Need to utilize the full scope of pharmacy practice |
| 3. Poor regulation of the pharmacy practice. | • Outdated regulatory policies on pharmacy practice.<br>• Insufficient monitoring of the practice of community pharmacists.<br>• Ethical concerns and conflicts-of-interest. |
| 4. Difficult practice environment | • Decreased profitability of community pharmacists.<br>• Undermining the role of pharmacy professional.<br>• Work overload and long shifts in the pharmacy profession. |

rather than limiting the number of students admitted into pharmacy schools as the latter might negatively affect the financial status of universities and the quality of education.

*"If you limit the number of graduates, will universities be able to continue paying for professors. . .I want to limit the numbers of pharmacists coming from abroad and maintain those in Lebanon."*

(*Community practitioner 2*)

**Public perception of pharmacy profession.** Interviewees mentioned a combination of factors that encouraged students to pursue pharmacy education, and leading to the oversupply of graduates. These factors include viewing pharmacy as a prestigious and profitable profession, with many benefits that could be earned from the OPL upon registration.

*"A lot of people are studying pharmacy because they have a misconception that a pharmacy makes a lot of money"*

(*Policy maker 4*)

*"The OPL has a very good retirement plan. . .and a good insurance. . .So it is attractive to working and retired pharmacists"*

(*Policy maker 9*)

## The demand-supply imbalance in the pharmacist workforce

**Need for additional investments in the pharmaceutical sector.** Interviewees noted that the inadequate investment in the various pharmaceutical sectors was widening the demand-supply gap. As such, they suggested creating more job opportunities in research, academia, and manufacturing.

*"Create new job opportunities in research. In Lebanon Pharmacists mainly work as medical representatives, in regulation, community, hospital. Outside Lebanon they work in research, local manufacturing, academia"*

(*Policy maker 6*)

Other interviewees elaborated that the development of the pharmacist educational programs was not matched with an advancement in pharmacy practice.

*"The pharmacy education has greatly advanced whereas. . .the practice of pharmacists has not changed much."*

(*Dean of pharmacy school 1*)

**Need to utilize the full scope of pharmacy practice.** Several interviewees acknowledged that the root of the problem was not the high number of graduates, but the misallocation of pharmacists in their proper roles. Interviewees added that the mismatch between pharmacy education and the actual practice was a contributing factor for their inability to find jobs. For example, many interviewees thought that pharmacists were not exercising their full scope of practice and were rather overqualified for the type of services they were providing.

*"There is lack of planning, lack of appreciation, limited professional development, all of this limits the pharmacists in their roles. . .Pharmacists have studied a lot but they are not permitted to implement what they have learned."*

(*Dean of pharmacy school 2*)

## Poor regulation of the pharmacy practice

**Outdated regulatory policies on pharmacy practice.**    According to interviewees, the poor regulation of the pharmacy profession in the country has led to several workforce challenges. The laws were either absent, or not being implemented, or needed updating. For example, according to law, opening a new pharmacy requires a 300 meters distance between the closest pharmacy, instead of the population number or the region needs, leading to numerous pharmacies opening in a certain area.

*"Today we have to get out of the idea that pharmacies should be 300 meters apart, to the idea of the need of the population for pharmacies. And this should be studied at the Ministry of Health."*

(*Community practitioner 2*)

Most of the interviewees attributed the lack of hospital pharmacists in the country to the absence of a law that specifies the number of hospital pharmacists per bed.

*"If a law is not present to mandate the presence of a certain number of pharmacists per a specific number of beds, hospitals will be the ones taking this decision"*

(*Hospital pharmacist 3*)

**Insufficient monitoring of the practice of community pharmacists.**    Another challenge in the community pharmacy sector was the absence of close supervision on pharmacist's practice and the implementation of laws by the relevant authorities. Interviewees articulated that some pharmacists were dispensing drugs without medical prescription or placing discounts on some of the products, thus, violating article 80 of the law on drug pricing.

*"In community pharmacies there are lots of drugs that are sold without a prescription for the purpose of making profit. . .the MOPH should have a bigger regulatory role"*

(*Health journalist*)

**Ethical concerns and conflicts-of-interest.**    Corruption was noted by interviewees as an aspect of the governmental authorities that led to the poor regulation of the pharmacy profession. One interviewee identified that conflicts-of-interest between members of different governmental bodies hampered the formulation of health-related laws.

*"Decisions are not based on evidence, but rather on financial considerations."*

(*Policy Maker 8*)

Multiple interviewees pointed out that in the absence of regulatory policies, the high number of pharmacies will lead to unethical practices or competition between community pharmacists.

*"Competition between pharmacies and illegal practices arise when the number of pharmacies exceeds the needs of the population, which is the case in some areas in Lebanon."*

(*Community practitioner 2*)

### Difficult practice environment

**Decreased profit in the community pharmacy sector: Implications on practice.**   Most of the interviewees referred to the reduction in the price of drugs as the major contributing factor to the decreased profit in community pharmacies. The price of drugs was reduced, which significantly decreased the profit margin of community pharmacists and hindered their ability to employ additional pharmacists as required by the Lebanese law. Interviewees emphasized that the low pharmacists to pharmacy ratio was not due to the poor law implementation, but it was due to the lack of profit in the sector itself.

*"The government lowered the prices without going back to the order of pharmacists. . .they should have increased the profit margin to compensate for the pharmacists' losses."*

(*Community practitioner 1*)

**Undermining the role of pharmacy professionals.**   Many interviewees elaborated that stakeholders in different sectors are playing a major role in underestimation of pharmacists and preventing them from widening their scope of practice to apply their professional skills as health care providers. Furthermore, the lack of collaboration between pharmacists and other healthcare professionals was raised by many interviewees as a reason behind undermining the profession.

*"Pharmacists in general do not have the chance to practice their full scope due to the perception of a conflict with other health care providers, which should not be the case if collaboration exists among health care providers"*

(*Professor of pharmacy 2*)

Interviewees discussed how the roles of community pharmacists were limited to the dispensing and selling of medications. They added that the pharmacists' scope of practice should be inclusive of other roles such as medical counselors, educators, and advocates.

*"A pharmacist's role is not in dispensing medications, it should include exchanging a medication with a cost-effective alternative- the generic to replace the brand."*

(*Policy maker 1*)

**Work overload and long shifts in the pharmacy profession.**   Interviewees raised the issue of long working hours and work overload in community pharmacies since usually only one pharmacist has to deal with multiple tasks. Consequently, some pharmacists reported feeling fatigued after working for long hours.

*"A pharmacist doing all the pharmacy work all through the day with no specific shift, will feel physically and psychologically fatigued"*

(*Policy maker 5*)

Hospital pharmacists were also affected by the heavy workload in the hospitals. As noted by the interviewees, this was mainly due to the limited number of hospital pharmacists who are not able to properly carry out the required tasks.

*"One pharmacist alone cannot be responsible for stock management, dispensation, and managing the pharmacy operations plus having a clinical role."*

(*Hospital pharmacist 3*)

## Discussion

In this study, stakeholders asserted that the pharmacy profession in Lebanon requires a concerted effort by concerned stakeholders to ensure workforce optimization and improve working conditions. The supply of pharmacists is poorly planned and regulated leading to a situation of oversupply. The demand for pharmacists, on the other hand, is focused on the community pharmacy sector at the expense of other sectors that contribute to the health and wellbeing of the population (e.g. clinical pharmacists, manufacturing, etc.). These factors, added to insufficient investment in the pharmaceutical sector, is widening the demand-supply gap in the pharmacy labor market in the country. Most interviewees expressed concern that the situation of the pharmacy profession calls for immediate interventions to protect it from further deterioration and to enable pharmacists to exercise their full scope of practice in serving the population.

Interviewees described an obvious oversupply of pharmacists, which is expected to exacerbate, as the number of new pharmacy graduates continues to surpass the number of available jobs. This is related to both internal and external factors. On one hand, the number of pharmacy schools and graduates in Lebanon is not optimized to the national needs. On the other hand, foreign trained pharmacists could get easily licensed to practice after passing oral and written examinations [4]. According to OPL, the majority of foreign trained pharmacists originated from neighboring Arab countries, Eastern European countries and Russia. Although the number of those pharmacists have decreased over the years, they still represent about 30% of active pharmacists. The easy integration of foreign trained pharmacists into the Lebanese health system was indeed a contributing factor in creating a surplus [7]. In contrast, most countries (including the neighboring ones in the Middle East Region) have strong regulatory requirements that highly restrict (or ban) the integration of foreign trained pharmacists into their labor markets. In addition, there is no agreement between Lebanon and other countries that enables pharmacists trained in Lebanon to practice outside. This can explain why only 8% of pharmacists trained in Lebanon are reporting practicing outside the country [7], despite the highly qualified and internationally accredited pharmacy programs in Lebanon. Over the years, the supply of pharmacists has exceeded the demand and regulatory requirements did not support the integration of the surplus into the system of other countries.

Perhaps more disconcerting, is the absence of a national health human resources forecasting model to advise on the current and future needs for pharmacists in Lebanon [4, 7]. This has contributed to the lack of coordination between the educational and practice sectors. Hence, a forecasting model is urgently needed to advise on the current and future needs of pharmacists in the various sectors of employment. Forecasting and analyzing future workforce trends require a comprehensive assessment, with consideration to the demographic characteristics of the workforce, training and education schemes, changes in pharmacist roles, and technological advancements in the profession [1]. The planning and forecasting exercise will not be an easy task in Lebanon as it requires the consolidation of data from various stakeholders

and a partnership with academic institutions and development partners to ensure the validity of the generated models. An additional challenge lies in the lack of accurate population statistics and the continuous movement of refugees to and from the country. Having said that, the forecasting models, even if they were less accurate, would be instrumental to guide the pharmacy workforce development in the future. This forecasting process should include representatives from the MOPH, OPL, academic and research institutions, regional and international agencies and various pharmacy practice sectors. The forecasting model has to be coupled with a regulatory framework to right-size the supply of pharmacists and set a strategic health human resources plan focused on the pharmacy profession.

One of the proposed solutions is to remedy workforce oversupply is to implement "numerus clausus", which designates the number of students admitted each year. This has been applied in several developed countries and proved to be effective in regulating workforce trends without negatively affecting patients or the profession [1]. Within the Lebanese context, capping admissions to pharmacy schools will be difficult to implement and would violate the very nature of the Lebanese free market educational sector. Furthermore, even if "numerus clausus" is successfully implemented, it would not be enough to rectify the imbalances in the market unless coupled with other policy options.

The experiences of other countries will be helpful on that front including: expanding the scope of practice of pharmacists [19], diversifying their specializations as per the needs of the market [4], facilitating the integration of the surplus of pharmacists into regional and international labor markets experiencing a shortage [20], and restricting the integration of foreign trained pharmacists in to the labor market. Again, bringing balance to the labor market would require a concerted effort of the various pharmacy sector stakeholders and the building of a consensus on a strategic direction and regulatory options.

However, capped university admissions and restricting the integration of foreign-trained pharmacists would not bring a quick respite to the practicing pharmacists in Lebanon. In fact, effective utilization of the current pharmacists' workforce represents a prompt solution for the country with long-term benefits [21]. Certainly, more efforts should be directed towards investing in various pharmaceutical sectors including research, academia, manufacturing, and clinical practices. For example, there is evidence that pharmacists in Lebanon were underutilized in the hospital sector. The ratio of hospital pharmacists to hospitals was 1.9 pharmacists, with pharmacists mainly practicing their dispensary role rather than the much needed clinical responsibilities [7]. Lebanon could follow the example of other countries in the region by mandating a minimum ration of clinical pharmacist to patients in Lebanese hospitals [22, 23].

As a matter of fact, regulating pharmacists supply and creating cross-sector ob opportunities require prompt governmental regulations [24]. Other countries have enabled pharmacists to exercise their full scope of practice, for example, community pharmacists in the UK have been involved in research [25, 26], health coaching [27], as well as patient assessments and population-based programs [28]. If this can be achieved successfully in Lebanon, the profession could be up-scaled with pharmacists practicing as patient educators and population health agents. With the fast-changing environment that is witnessed globally, it is imperative to ensure a nimble, dynamic and evidence based regulatory process. Therefore, any intervention should be coupled with ongoing monitoring and periodic review of supply and demand in order to capture unintended consequences and address them in a timely manner.

Additionally, interviewees reported that the weak monitoring and evaluation of the implementation of existing laws and guidelines is hindering the development of the workforce. For instance, the regulatory policies that mandate the presence of pharmacists in their pharmacies throughout the operating hours is poorly enforced. This is undermining the role of pharmacists as medicine and drug experts. Interviewees further report unethical or illegal practices,

including: switching prescribed drugs (without the consent of the prescriber), offering discounts to lure customers, promoting drugs and products based on incentives offered by companies and distributors, and dispensing drugs without a prescription. As such, adequate monitoring of pharmacy practice is essential to ensure the safety and quality of the delivered services [29]. Moreover, interviewees raised concern with the lack of joint governance which leads to negative spillover effects on the profession of some of the governmental policies (e.g. reduction on drug prices without revising the business model of pharmacists).

Literature emphasize that good governance is key for achieving good pharmacy practice [29, 30]. Many interviewees described a general lack of trust in the role of the governing bodies due to corruption, incompetence and conflicts-of-interest. This constitutes a major barrier towards enforcing regulatory policies. Reinforcing trust is a prerequisite to successful and sustainable policy implementation that can improve pharmacy practice [30]. Joint governance is essential for maintaining high standards of care and improving quality of services [31]. The OPL efforts to rectify the labor market anomalies have not been successful with the government. For example, although the health committee of the Lebanese parliament has accepted the suggestions from OPL regarding implementation of clinical pharmacists in hospitals (accepted in September 2018) and the limitation on the number of pharmacy graduates (accepted in March 2019), the two laws have yet to be implemented [32].

This study identified other challenges for pharmacy practice in Lebanon, including lack of appreciation of pharmacists' role, poor interprofessional collaboration, and poor working conditions. Despite the significant changes that the pharmacy profession has witnessed worldwide, the poor recognition of pharmacists is still common in developing countries. Furthermore, the lack of collaboration between pharmacists and other healthcare professionals, especially physicians, was raised by many interviewees as a reason behind undermining the profession [33, 34]. Literature emphasize the importance of having open communication between healthcare professionals and respect for each other's roles [35]. Inter-professional collaboration has been reported in several studies as a driver for advanced pharmacy practice. It is a recognized benefit for pharmacists and a desirable component of patient-centered care, leading to better patient outcomes [36]. Hence, it is essential to articulate the importance of pharmacists as the medicinal and drug experts. This could be achieved through integrating pharmacists in outreach programs and national campaigns promoted by OPL and the MOPH.

Furthermore, study findings identified poor working conditions, such as long working hours and work overload. Given the increased competition between community pharmacies and the inability of pharmacy owners to hire pharmacy assistants, community pharmacists have to work for long hours [31]. Similar workplaces stressors have been reported in several countries such as the United Kingdom, Canada, and the United States [37]. These issues raise several concerns regarding pharmacists' well-being and patient safety. Thus, working conditions must be monitored and improved by relevant authorities. Stakeholders must ensure that the workforce is adequately staffed, qualified, and utilized to complete the required tasks. They must prioritize pharmacists' physical, mental, and psychological well-being. Additionally, employers and schools of pharmacy should establish support programs with various resources and strategies that can improve the well-being and resilience of pharmacists [38].

## Limitations

A number of shortcomings in this study warrant mentioning. First, the interviewed stakeholders may not represent the views of all experts involved in the pharmacy practice. Yet, the views expressed by interviewees suggest a shared understanding among all stakeholders. Second, the authors cannot rule out stakeholders' own impressions and biases. However, the research team

minimized the biases through identifying recurrent themes and by asking interviewees to provide further elaboration/evidence for unsubstantiated comments. In addition, the authors ensured that the questions in the interview guide are simple, non-leading, with minimal wording bias to avoid influencing the responses of the interviewees. Finally, the study could benefit from a stronger voice for the civil society. Regretfully, the country does not have civil society organizations/NGOs with the required level of knowledge in the field. The experienced opinion of impartial academicians and the media was utilized as a proxy to the voice of the civil society.

## Conclusion

The current situation in the pharmacy labor market in Lebanon is detrimental to the financial and professional wellbeing of Lebanese pharmacists and may jeopardize the quality and safety of service extended to the population. A concerted effort, led by the MOPH and OPL, and involving the stakeholders, is necessary to enhance workforce planning, regulate supply, optimize the integration of pharmacists into work sectors of need and to enhance the financial and professional wellbeing of pharmacists in Lebanon. Such efforts, if successful, could transform a profession in danger into a major contributor to the health and wellbeing of individuals and communities.

## Acknowledgments

The authors wish to extend deep gratitude to all the interviewees who took time off their busy schedule to contribute to this study. The authors are also indebted to the Order of Pharmacists in Lebanon for supporting this study.

## Author Contributions

**Conceptualization:** Mohamad Alameddine, Mohamad Ali Hijazi.

**Data curation:** Mohamad Alameddine, Sara Kassas.

**Formal analysis:** Mohamad Alameddine, Sara Kassas, Mohamad Ali Hijazi.

**Funding acquisition:** Mohamad Alameddine.

**Investigation:** Mohamad Alameddine, Karen Bou-Karroum, Mohamad Ali Hijazi.

**Methodology:** Mohamad Alameddine, Mohamad Ali Hijazi.

**Project administration:** Mohamad Alameddine, Karen Bou-Karroum, Mohamad Ali Hijazi.

**Supervision:** Mohamad Alameddine, Mohamad Ali Hijazi.

**Validation:** Mohamad Alameddine, Sara Kassas, Mohamad Ali Hijazi.

**Visualization:** Karen Bou-Karroum.

**Writing – original draft:** Mohamad Alameddine, Karen Bou-Karroum, Sara Kassas, Mohamad Ali Hijazi.

**Writing – review & editing:** Mohamad Alameddine, Karen Bou-Karroum, Mohamad Ali Hijazi.

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
