## [Decision Letter · Decision Letter 0]

1 Oct 2020

PONE-D-20-28091

A profession in danger: Stakeholders’ perspectives on supporting the pharmacy profession in Lebanon

PLOS ONE

Dear Dr. Hijazi,

Thank you for submitting your manuscript to PLOS ONE. After careful consideration, we feel that it has merit but does not fully meet PLOS ONE’s publication criteria as it currently stands. Therefore, we invite you to submit a revised version of the manuscript that addresses the points raised during the review process.

We look forward to receiving your revised manuscript.

Kind regards,

Vijayaprakash Suppiah, PhD

Academic Editor

PLOS ONE

Journal Requirements:

2. Please include additional information regarding the qualitative interview guide used in the study and ensure that you have provided sufficient details that others could replicate the analyses. For instance, if you developed an interview guide as part of this study and it is not under a copyright more restrictive than CC-BY, please include a copy, in both the original language and English, as Supporting Information.

3. In the Methods, please discuss whether and how the questionnaire was validated and/or pre-tested. If this did not occur, please provide the rationale for not doing so.

Reviewers' comments:

Reviewer's Responses to Questions

**Comments to the Author**

1. Is the manuscript technically sound, and do the data support the conclusions?

Reviewer #1: Partly

Reviewer #2: Yes

2. Has the statistical analysis been performed appropriately and rigorously? 

Reviewer #1: N/A

Reviewer #2: N/A

3. Have the authors made all data underlying the findings in their manuscript fully available?

Reviewer #1: No

Reviewer #2: Yes

4. Is the manuscript presented in an intelligible fashion and written in standard English?

Reviewer #1: Yes

Reviewer #2: Yes

5. Review Comments to the Author

Reviewer #1: I think this is an interesting paper. The surplus of pharmacists is a troubling issue in many countries, including but not limited to Lebanon. This issue - especially the details of the mechanism by which market and policy forces may alleviate this surplus - have not been fully explored in the literature.

Being a purely qualitative study, the transparency of the study's details are especially crucial in determining whether the study is of sufficient technical and rigorous merit to warrant publication. Many of these details are missing. I will note a few major issues below, noting that the list is not exhaustive.

1. What is the distribution of degrees awarded between the B.S. and the Pharm.D. degree each year? Are these scope of practice differences? Is the PharmD exclusively intended to work in institutional or other advanced practice settings? We need to know whether the surplus of pharmacists is a function of structural academic characteristics before it is possible to interpret the results of your analysis.We also need to know more about the specific clinical roles of a pharmacist - what is in place currently versus the aspirations of the profession as a whole.

2. The study also consistently posits a clausus as a means to address the surplus. Would such a clausus violate antitrust or anti-collusion laws in Lebanon? Are there alternative policy or market responses that could be considered as viable responses? By including this discussion in the introduction (in the manner in which it is introduced) it potentially biases the reader towards only one of many responses.

3. On page 6, why would robberies contribute to the over-supply of pharmacists? Would it not lead to the opposite effect? Also, the authors note that pharmacists trained outside of Lebanon contribute to the surplus of pharmacists. From where do the primarily originate? Do Lebanese trained pharmacists have the ability to practice internationally? If so, where can they practice? One cannot discuss one side of the labor market without addressing the other.

4. More details are necessary to justify the research design. How did you arrive at a number of 21 stakeholders? There are many details and considerations beyond simply describing "purposive sampling" that must be described to ensure the study's accuracy and transparency. Why not 30? Why not 12? Holding constant the number of stakeholders, please explain why the distribution of the expertise and practice settings of the 21 stakeholders is sufficient to justify accurate, reliable and precise themes? Are they set proportionally to the distribution of practicing pharmacists? Or is there some other rationale to deviate from this proportionality? It seems the respondents are ore heavily weighted towards policy than towards other areas, most notably community practice (where most of the surplus is occurring). For your policy maker stakeholders, what is the scope of their work related to policy making? What do they do?

5. Did anyone decline to participate as a stakeholder? Were there people excluded from participating, and on what grounds?

6. The most glaring limitation of the study is that this reviewer was not able to view the actual interview questions and structure of the interview (preface, conclusion, etc.). The details of the wording of the interview questions (including their scope of coverage, but also the neutrality or slanting of the wording of the questions) is crucial in generating information and ultimately the creation of themes. The second paragraph on page 7 should therefore be expanded substantially to include these details.

7. What was the process (which should have been identified a priori to the study's implementation) by which possible disagreements in the creation of the themes were resolved?

8. Please better and more precisely explain what you mean by "A preliminary data analysis was performed in order to identify issues that need further investigation." This is highly ambiguous and be interpreted in many ways, some of which may undercut the legitimacy of the research. I am not saying this did happen, but it should be clarified to covey to a reader that it did not happen.

9. Please explain in more detail your coding process, what words and expressions were used to identify the key concepts. Again, this is the "art" of qualitative research, and it should be described in more detail so the readers know exactly what you are trying to accomplish.

10. The second "Contributing Factors to the Oversupply" of Pharmacists in Lebanon" is superficial and needs much greater development (greater depth of analysis), as well as tied to the literature. I think what is contained in this section is fine as a starting point. But there is really no information in each subsection that would not be intuitively obvious to an individual with (even rudimentary) knowledge of the industry. It is important to not only summarize the responses, but to use them to identify novel research questions or to identify previously undiscussed policy issues. For example, the first idea and subsection discusses the clauses concept identified in the introduction. OK; certain countries use clauses to restrict supply. Is the Lebanese educational system designed to enforce clauses? What would it take to enact a clausus in terms of reform to the pharmacy educational system? Was any stakeholder able to identify what the correct number of students admitted to pharmacy school is or approximately should be?

What is reported is too superficial to be of use to identifying research questions or specific directions for future policy research, which is the goal of any qualitative study.

Similar types of issues pervade the remainder of this section. Since this section touches on many major themes, it is infeasible to address points of detail for each of these sections, except to note that each point, while entirely valid in premise, should be expanded with depth of analysis to ensure that the manuscript appropriately and consistently accomplishes its objectives.

11. In the Conclusions section you mention the creation of a forecasting model. I think this is a good idea. However, there are two issues that should be addressed here. First, a model should forecast with a purpose, and include forecasts that anticipate or predict the effects of specific policy responses. Other than simply forecasting graduates and current openings, what should the model specifically address as the most pressing issue? Is it the clauses, expanded scope of practice, etc.? Second, forecasting itself is not a political process. The model is built to forecast phenomena based on the needs of policy holders. The results are used to inform policy actions. But the model itself should be designed and implemented based on rigorous scientific standards, not on the consensus of politicians or stakeholders.

Reviewer #2: The article is well written and establishes data to support its main points. The lack of a validated interview tool may be an important weakness to point out. The summary and analysis of the data in the results and discussion sections are clear and well explained.

It is challenging to remove writer, interviewer and interviewee bias from a manuscript like this. The authors attempt to reduce this and provide information in the limitation section. Perhaps even more emphasis of this point should be attempted in the limitation section.

More support could have been provided to the information about limitation of pharmacists from foreign graduates getting licensed could have been offered. Additionally, consideration of the impact/problems/issues that may result from overall more regulation should have been discussed further.

Thanks for this contribution to the literature and effort to undertake such an extensive interview process with stakeholders.

6. PLOS authors have the option to publish the peer review history of their article (what does this mean?). If published, this will include your full peer review and any attached files.

Reviewer #1: No

Reviewer #2: No

---

## [Author Response · Author response to Decision Letter 0]

22 Oct 2020

Comments from Reviewer #1

The authors would like to extend their deep gratitude to the reviewer for the helpful feedback. We are delighted you found this article interesting and concur with you that the topic addressed by this manuscript requires further attention in literature. We found great value in your suggested edits and will use them as a guide to improve the writeup, comprehensiveness and clarity of this manuscript. 

Please see below a point-by-point response to your comments and suggestions. 

Comment 1.1: What is the distribution of degrees awarded between the B.S. and the Pharm.D. degree each year? Are these scope of practice differences? Is the PharmD exclusively intended to work in institutional or other advanced practice settings? We need to know whether the surplus of pharmacists is a function of structural academic characteristics before it is possible to interpret the results of your analysis. We also need to know more about the specific clinical roles of a pharmacist – what is in place currently versus the aspirations of the profession as a whole.

Response: We are happy to provide additional clarification on the distribution and professional scope of pharmacists in Lebanon. The country currently has five active universities offering pharmacy programs (4 private, 1 public). Pharmacy students can graduate with a bachelor’s in pharmacy degree (5-year pharmacy program) or a PharmD degree (6-year pharmacy program including clinical rotations). Due to the absence of a national plan guiding the supply and utilization of pharmacists, there is no accurate database on the exact distribution of degrees awarded between the B.S. and the Pharm.D.. Moreover, the number of Pharm.D. graduates fluctuates annually and depend on students' desire to pursue the program and on a university's capacity to offer the degree. Pharmacists with Pharm.D. degrees should in principle have an advantage over B.S. holders with respect to clinical practice. They are trained to work at hospitals as clinical pharmacists, on the floor, as consultants or in managed care. The clinical pharmacist’s role is to make sure patients understand and adhere to their medication regimen with respect to timing, dosage, and frequency during the prescribed length of time. In addition, the clinical pharmacist is trained to prevent potential adverse drug events by providing drug information in plain language to the patient at key transition times, such as hospital admission and discharge, as well as during the patient’s entire stay (Ghaibi et al., 2015). However, the practice of clinical pharmacists in Lebanon tells a different story, with a recent study revealing that they are primarily practicing their drug dispensing role rather than their clinical role (Alameddine et al., 2019). The average ratio of hospital pharmacists to hospitals was 1.9 in Lebanon, with an unfortunate absence of pharmacists from clinical floors or divisions. The hospital pharmacist’s role in Lebanon focuses on the distribution of medications, compounding, formulary review, and cost management. The aspirations for pharmacy stakeholders is for clinical/hospital pharmacists to exercise their full scope of practice, including: ensuring safe, appropriate, efficacious, judicious, and cost-effective use of medications while decreasing improper medication usage and enhancing the satisfaction of patients (Dalton & Byrne, 2017; Vaillancourt, 2011). 

Comment 1.2: The study also consistently posits a clausus as a means to address the surplus. Would such a clausus violate antitrust or anti-collusion laws in Lebanon? Are there alternative policy or market responses that could be considered as viable responses? By including this discussion in the introduction (in the manner in which it is introduced) it potentially biases the reader towards only one of many responses.

Response: The reviewer makes a good point here. There are no antitrust or anti-collusion laws in Lebanon. Therefore, limiting the number of students does not violate antitrust laws. On the contrary, according to the Order of Pharmacists (OPL) and other experts in the field, limiting the number of graduates could potentially enhance the quality of pharmacy education. The clauses would be aligned with community needs and would not affect the competition between academic institutions. 

Having said that, we concur with the reviewer that the suggested “clauses” is not the only policy option and it would not be enough if implemented on its own. Health authorities, education authorities and the OPL should work cooperatively to make the best use of pharmacy graduates and limit their surplus. Several suggestions were raised including the integration of more specialization in pharmacy education such as: pharmaceutical industries, research on drug discovery, consultant, care management, etc (Hallit et al., 2019). In addition, there is a necessity to expand the scope of practice and issue the law of clinical pharmacists (accepted by Parliamentary Health Committee on September 2018) that will utilize hundreds of qualified pharmacists that are Pharm.D holders or clinically certified (Brown, 2013). Another suggestion for policy makers is to create bilateral/multilateral agreements to facilitate the employment of the surplus of Lebanese pharmacists in countries that suffer from a shortage of pharmacists (OECD, 2004). FIP reported that pharmacists, in particular, are lacking in the workforce in many countries like those in Africa, South East Asian, and the Middle East (Bates et al., 2018). These options are mentioned in the manuscript on page 19-20. 

Comment 1.3: On page 6, why would robberies contribute to the over-supply of pharmacists? Would it not lead to the opposite effect? Also, the authors note that pharmacists trained outside of Lebanon contribute to the surplus of pharmacists. From where do they primarily originate? Do Lebanese trained pharmacists have the ability to practice internationally? If so, where can they practice? One cannot discuss one side of the labor market without addressing the other.

Response: Robberies were listed under the challenges facing community pharmacists and thus it does not contribute to the oversupply of pharmacists. In fact, the increasing frequency of robberies is contributing to the burnout of pharmacists and is creating a risky work environment. We have clarified the point in the manuscript.

According to OPL, the majority of foreign trained pharmacists originated from neighboring Arab countries, Eastern European countries and Russia. Although the number of those pharmacists have decreased over the years, they still represent about 30% of active pharmacists. The easy integration of foreign trained pharmacists into Lebanese health system was indeed a contributing factor in creating a surplus (Alameddine et al., 2019). In contrast, most countries (including the neighboring ones in the Middle East Region) have strong regulatory requirements that highly restrict (or ban) the integration of foreign trained pharmacists into their labor markets. In addition, there is no agreement between Lebanon and other countries that enables pharmacists trained in Lebanon to practice outside. This could explain why only 8% of pharmacists trained in Lebanon are reporting practicing outside the country (Alameddine et al., 2019), despite the highly qualified and internationally accredited pharmacy programs in Lebanon. Over the years, the supply of pharmacists has exceeded the demand and regulatory requirements did not support the integration of the surplus into the system of other countries. This has been elaborated in the discussion in the manuscript (page 18).

Comment 1.4: More details are necessary to justify the research design. How did you arrive at a number of 21 stakeholders? There are many details and considerations beyond simply describing "purposive sampling" that must be described to ensure the study's accuracy and transparency. Why not 30? Why not 12? Holding constant the number of stakeholders, please explain why the distribution of the expertise and practice settings of the 21 stakeholders is sufficient to justify accurate, reliable and precise themes? Are they set proportionally to the distribution of practicing pharmacists? Or is there some other rationale to deviate from this proportionality? It seems the respondents are heavily weighted towards policy than towards other areas, most notably community practice (where most of the surplus is occurring). For your policy maker stakeholders, what is the scope of their work related to policy making? What do they do? 

Response: We thank the reviewer for the valuable comment and appreciate the opportunity to further elaborate on our study design and methods. The number of interviews carried out in this study was not predetermined prior to the start of the study (our apologies if we were not clear on this point in the initial submission), but rather emerged as the authors reached the saturation point and the themes and subthemes became redundant (Strauss, 1998). We have explained our sampling process in more detail in the manuscript on page 6-7. 

The Lebanese market is relatively small and stakeholders who can contribute to the profession are well-known and were all included in our sampling frame. As for academicians and practicing pharmacists, we had to sample those who are opinion leaders with lots of experience. The 21 stakeholders we ended up interviewing were from different regions and institutions in the country. 

The Lebanese regulation allows policy makers at different levels (OPL, parliament, Ministry of health) to continue their community services at their own pharmacies or other institutions. Although most interviewed pharmacists were policy makers, most of them have many years of experience as community pharmacists. The participant list was chosen to represent all the Governorate of Lebanon. The experience of interviewed policy makers in the pharmacy labor market is represented below. Similarly, all other stakeholders have experience in the pharmacy labor market. The authors are ready to share the experience of all interviewees upon the reviewer’s request.

Interviewee Practice experience Policy experience Region

P1 Owner of Pharmaceutical company for more than 30 years

Owner of pharma drugstore for more than 20 years Current President of OPL

Former president of OPL (2003-2006; 1993-1996)

Member of OPL board for several years South Lebanon

P2, P3, P5 Owner of a community pharmacies for more than 20 years Former Presidents of OPL Beirut Governorate

Mount Lebanon Governorate

P5 Owner of community pharmacy for more than 25 years Active trustee of OPL North Lebanon

P6 More than 10 years experience in pharma companies Head of Pharmacy department at Ministry of Public Health Beirut Governorate

P7 Owner of Pharmacy for more than 15 years

10 years experience in pharmaceutical industry Health minister consultant for pharmaceutical issues South Lebanon

As is evident in the above Table, the various Governorates of Lebanon were represented. Having said that, and due to the small size of the country and the centralization of regulatory framework and policy making, the themes emerging from various stakeholders converged. In this study, the authors were more concerned with ensuring the thorough expertise of interviewees more than their regional representation.

 Comment 1.5: Did anyone decline to participate as a stakeholder? Were there people excluded from participating, and on what grounds?

Response: Thanks for the opportunity to clarify this point. The research team was very flexible and waited until the stakeholders were available and scheduled the meetings at their convenience. This has led to a high acceptance rate. All of the invited stakeholders except for one agreed to participate in this study. The main reason for non-participation was the lack of time. As such, an experienced stakeholder from the same institution was nominated. 

Comment 1.6: The most glaring limitation of the study is that this reviewer was not able to view the actual interview questions and structure of the interview (preface, conclusion, etc.). The details of the wording of the interview questions (including their scope of coverage, but also the neutrality or slanting of the wording of the questions) is crucial in generating information and ultimately the creation of themes. The second paragraph on page 7 should therefore be expanded substantially to include these details.

Response: As suggested by the reviewer, kindly find below a copy of the interview guide. We have integrated the interview guide in the methodology section in the manuscript on page 8-9.

1- Please describe your experiences in the pharmacy sector in Lebanon? 

2- What is your opinion on the labor market of Lebanese pharmacists?

- What needs to be improved?

- Why do you think this is the case?

3- What necessary measures need to be implemented to ensure future balance in the pharmacy labor market in Lebanon?

- Why would these measures be important?

- How could they be implemented?

4- What are the requirements to upscale the role of pharmacist in Lebanon?

- What are the impediments? 

- What needs to be done to overcome them?

- Who are the responsible parties? 

5- How do you describe the practice environment of pharmacists in Lebanon?

- Why do you think this is the case?

- How do you think this could be improved?

6- Are there any additional issues or improvements you wish to highlight?

7- Who else should we speak with to help answer our questions?

Comment 1.7: What was the process (which should have been identified a priori to the study's implementation) by which possible disagreements in the creation of the themes were resolved?

Response: We are happy to elaborate on this point. It is common for different research members to disagree about the appropriateness of codes used for labeling datasets in qualitative analysis. In our study, two members of the research team were responsible for coding and met regularly to focus on identifying points of probable inconsistency and resolve disagreements. Disagreements that could not be resolved were referred to a third team member; however, consensus was reached very often. We have added this process to the manuscript on page 9. 

Comment 1.8: Please better and more precisely explain what you mean by "A preliminary data analysis was performed in order to identify issues that need further investigation." This is highly ambiguous and be interpreted in many ways, some of which may undercut the legitimacy of the research. I am not saying this did happen, but it should be clarified to covey to a reader that it did not happen.

Response: We apologize for being unclear, and we thank you for giving us the opportunity to interpret this point. During the data analysis phase, transcribed interviews were comprehensively read in order to gain preliminary insight into the data. The aim of this early stage of thematic analysis is to get researchers familiarized with the responses of the interviews and sensitized to early themes. We have clarified this point in the manuscript on page 9. 

Comment 1.9: Please explain in more detail your coding process, what words and expressions were used to identify the key concepts. Again, this is the "art" of qualitative research, and it should be described in more detail so the readers know exactly what you are trying to accomplish.

Response: Thank you for the suggestion. Transcribed interviews were imported to Dedoose, a qualitative analysis software (Lieber, 2014). We used the grounded theory approach in our analysis, a process of coding that uses phrases and terms used by interviewees for developing codes (Khan, 2014). Two members of the research team completed inductively, line-by-line coding for the transcripts without the use of a priori coding scheme. Codes were attached to excerpts, and an initial set of codes was developed from the recurring ideas and discussion points. Multiple rounds of co-coding were applied until consensus on codes definitions was reached. Duplicate codes were removed and overlapping codes were merged. Then, the identified set of codes was explored to form sub-themes from the conceptually related codes. Finally, sub-themes were grouped to form overarching themes. The themes were applied to each transcript, with amendments made if new codes emerged. The final list of themes and sub-themes was approved by all research team members. We have explained the coding process in more detail on page 9 in the manuscript. 

Comment 1.10: The second "Contributing Factors to the Oversupply" of Pharmacists in Lebanon" is superficial and needs much greater development (greater depth of analysis), as well as tied to the literature. I think what is contained in this section is fine as a starting point. But there is really no information in each subsection that would not be intuitively obvious to an individual with (even rudimentary) knowledge of the industry. It is important to not only summarize the responses, but to use them to identify novel research questions or to identify previously undiscussed policy issues. For example, the first idea and subsection discusses the clauses concept identified in the introduction. OK; certain countries use clauses to restrict supply. Is the Lebanese educational system designed to enforce clauses? What would it take to enact a clausus in terms of reform to the pharmacy educational system? Was any stakeholder able to identify what the correct number of students admitted to pharmacy school is or approximately should be?

What is reported is too superficial to be of use to identifying research questions or specific directions for future policy research, which is the goal of any qualitative study.

Similar types of issues pervade the remainder of this section. Since this section touches on many major themes, it is infeasible to address points of detail for each of these sections, except to note that each point, while entirely valid in premise, should be expanded with depth of analysis to ensure that the manuscript appropriately and consistently accomplishes its objectives.

Response: We thank the reviewer for the valuable feedback, the team have extensively revised the discussion section to ensure contextualization of all policy options while maintaining a proper word limit. Additional input was added throughout the discussion section on pages 19 and 20. The authors acknowledge that numerus clausus would be difficult to implement knowing the nature of the Lebanese free market educational sector. They also acknowledge that numerus clausus alone would not be enough to bring balance to the labor market and should be coupled with other policy options. As such, other solutions to regulate the pharmacists supply and demand were mentioned in the discussion, including expanding pharmacists’ scope of practice, facilitating the integration of pharmacists in labor markets of countries experiencing shortages, and creating cross-sector job opportunities. All the discussed policy options require concerted efforts from various stakeholders and academic institutions to ensure proper implementation. Similarly, forecasting and analyzing future workforce trends require comprehensive assessment and accurate data to ensure the validity of the generated models. Though it would be challenging to obtain accurate population statistics, the forecasting model is necessary to guide the pharmacy workforce development in the future. 

Comment 1.11: In the Conclusions section you mention the creation of a forecasting model. I think this is a good idea. However, there are two issues that should be addressed here. First, a model should forecast with a purpose, and include forecasts that anticipate or predict the effects of specific policy responses. Other than simply forecasting graduates and current openings, what should the model specifically address as the most pressing issue? Is it the clauses, expanded scope of practice, etc.? Second, forecasting itself is not a political process. The model is built to forecast phenomena based on the needs of policy holders. The results are used to inform policy actions. But the model itself should be designed and implemented based on rigorous scientific standards, not on the consensus of politicians or stakeholders.

Response: Thank you for your comment, further input into the generation of the forecasting model within the Lebanese context have been added, please check page 19 in the manuscript. The main purpose of the forecasting model is to reflect on the adequacy of current number of pharmacists to address the needs of the population. It also provides advice on the future anticipated number of pharmacists. On that front, the authors believe that the forecasting model would be more likely guiding policy formulation to address market disbalances and optimize the use of pharmacists in serving individuals and communities. We strongly concur with the reviewer that for this model to be of utility, has to be based on rigorous scientific standards and be shielded to the extent possible from the interference of politicians and biases of stakeholders. This is why the authors call for the engagement of experts from academic institutions and international agencies to safeguard the integrity of the process and the validity of the findings.

Comments from Reviewer #2

The authors would like to extend their deep gratitude to reviewer #2 for the helpful feedback. We found great value in your suggested edits and will use them as a guide to improve the writeup, comprehensiveness and clarity of this manuscript. 

Please see below a point-by-point response to your comments and suggestions. 

Comment 2.1: The article is well written and establishes data to support its main points. The lack of a validated interview tool may be an important weakness to point out. The summary and analysis of the data in the results and discussion sections are clear and well explained.

Response: Thank you for providing your valuable feedback on the manuscript. We are sorry for missing the inclusion of the interview guide in the first submission. We have now integrated it in the methodology section in the manuscript on page 8-9. 

Comment 2.2: It is challenging to remove writer, interviewer and interviewee bias from a manuscript like this. The authors attempt to reduce this and provide information in the limitation section. Perhaps even more emphasis of this point should be attempted in the limitation section.

Response: We are happy to provide additional information on the authors’ attempt to reduce biases in this qualitative study. In this study, researchers avoided pre-existing assumptions and interpretations that support existing beliefs and hypotheses. On the same note, the authors did their best to keep the questions in the interview schedule simple, non-leading, with minimal wording bias to avoid influencing the responses of the interviewees. This point was added to the limitations section in the manuscript (page 23). 

Comment 2.3: More support could have been provided to the information about limitation of pharmacists from foreign graduates getting licensed could have been offered. Additionally, consideration of the impact/problems/issues that may result from overall more regulation should have been discussed further.

Response: Thank you for pointing this out. As suggested by the reviewer, we added more information on the integration of foreign trained pharmacists (page 18 in the manuscript). The challenge here is that foreign trained pharmacists can easily practice in Lebanon after passing an exam. However, the surplus of pharmacists is not easily integrated into the labor markets of other countries. This is due to the absence of an agreement between Lebanon and other countries that allows pharmacists trained in Lebanon to practice outside. It is worth noting that foreign trained pharmacists represent 30% of currently practicing pharmacists in Lebanon, whereas only 8% of pharmacists trained in Lebanon are practicing outside (Alameddine et al., 2019).

While we call on the stakeholders to enforce stricter evidence-based regulations to the market, we wish to stress that this process should always be dynamic and guided by evidence rather than the whims of politicians or the biases of policy makers. With the fast-changing environment that is witnessed globally, it is imperative to ensure a nimble, dynamic and evidence based regulatory process. Therefore, any intervention should be coupled with ongoing monitoring and periodic review of supply and demand in order to capture unintended consequences and address them in a timely manner. This point has been added to the discussion in the manuscript (page 20-21). 

 

References 

Alameddine, M., Karroum, K. B., & Hijazi, M. A. (2019). Upscaling the pharmacy profession in Lebanon: workforce distribution and key improvement opportunities. Human Resources for Health, 17(1), 47. 

Bates, I., John, C., Seegobin, P., & Bruno, A. (2018). An analysis of the global pharmacy workforce capacity trends from 2006 to 2012. Human Resources for Health, 16(1), 1-9. 

Brown, D. L. (2013). A looming joblessness crisis for new pharmacy graduates and the implications it holds for the academy. American journal of pharmaceutical education, 77(5). 

Dalton, K., & Byrne, S. (2017). Role of the pharmacist in reducing healthcare costs: current insights. Integrated pharmacy research & practice, 6, 37. 

Ghaibi, S., Ipema, H., & Gabay, M. (2015). ASHP guidelines on the pharmacist’s role in providing drug information. American Journal of Health-System Pharmacy, 72(7), 573-577. 

Hallit, S., Sacre, H., Hajj, A., Sili, G., Zeenny, R. M., & Salameh, P. (2019). Projecting the future size of the Lebanese pharmacy workforce: forecasts until the year 2050. International Journal of Pharmacy Practice, 27(6), 582-588. 

Khan, S. N. (2014). Qualitative research method: Grounded theory. International Journal of Business and Management, 9(11), 224-233. 

Lieber, E. (2014). Dedoose (Beginner Workshop)—Getting Started with Your Qualitative and Mixed Methods Data Management, Analysis, and Presentation. 

OECD. (2004). Migration for Employment: Bilateral Agreements at a crossroads. OECD Publishing. 

Strauss, A. (1998). 8c Corbin, J.(1998). Basics of qualitative research: Techniques and procedures for developing grounded theory. Thousand Oaks, CA: Sage. 

Vaillancourt, R. (2011). Pharmacists: the guardians of safe medication use. The Canadian Journal of Hospital Pharmacy, 64(1), 5.

---

## [Decision Letter · Decision Letter 1]

29 Oct 2020

A profession in danger: Stakeholders’ perspectives on supporting the pharmacy profession in Lebanon

PONE-D-20-28091R1

Dear Dr. Hijazi,

We’re pleased to inform you that your manuscript has been judged scientifically suitable for publication and will be formally accepted for publication once it meets all outstanding technical requirements.

Kind regards,

Vijayaprakash Suppiah, PhD

Academic Editor

PLOS ONE

Additional Editor Comments (optional):

Reviewers' comments:

Reviewer's Responses to Questions

**Comments to the Author**

1. If the authors have adequately addressed your comments raised in a previous round of review and you feel that this manuscript is now acceptable for publication, you may indicate that here to bypass the “Comments to the Author” section, enter your conflict of interest statement in the “Confidential to Editor” section, and submit your "Accept" recommendation.

Reviewer #1: All comments have been addressed

Reviewer #2: All comments have been addressed

2. Is the manuscript technically sound, and do the data support the conclusions?

Reviewer #1: Yes

Reviewer #2: Yes

3. Has the statistical analysis been performed appropriately and rigorously? 

Reviewer #1: Yes

Reviewer #2: Yes

4. Have the authors made all data underlying the findings in their manuscript fully available?

Reviewer #1: Yes

Reviewer #2: Yes

5. Is the manuscript presented in an intelligible fashion and written in standard English?

Reviewer #1: Yes

Reviewer #2: Yes

6. Review Comments to the Author

Reviewer #1: The authors have made a very thorough and meaningful attempt to address my comments. Most of my major concerns with the manuscript have now been addressed. I comment the authors for their efforts to improve the manuscript.

Reviewer #2: All of my areas of recommendation have been addressed from my first review. I support the edits and responses received to my comments.

7. PLOS authors have the option to publish the peer review history of their article (what does this mean?). If published, this will include your full peer review and any attached files.

Reviewer #1: No

Reviewer #2: No

---

## [Editor Report · Acceptance letter]

3 Nov 2020

PONE-D-20-28091R1 

A profession in danger: Stakeholders’ perspectives on supporting the pharmacy profession in Lebanon 

Dear Dr. Hijazi:

I'm pleased to inform you that your manuscript has been deemed suitable for publication in PLOS ONE. Congratulations! Your manuscript is now with our production department. 

Kind regards, 

on behalf of

Dr. Vijayaprakash Suppiah 

Academic Editor

PLOS ONE